# The impact of alcohol misuse in trauma patients: A scoping review protocol

**Chantae Garland**[1], **Nhayan Abdulla**[1,2], **Donghyun Lee**[1], **Rae Spiwak**[1], **Sarvesh Logsetty**[1], **Jordan Nantais**[1,3,4] *

**1** Department of Surgery, Max Rady College of Medicine, University of Manitoba, Winnipeg, Manitoba, Canada, **2** Department of Surgery, McGill University, Montreal, Quebec, Canada, **3** Li Ka Shing Knowledge Institute, St Michael's Hospital, Toronto, Ontario, Canada, **4** Institute of Medical Science, Temerty Faculty of Medicine, University of Toronto, Toronto, Ontario, Canada

* Jordan.nantais@mail.utoronto.ca

## Abstract

### Background

Alcohol use is a contributing factor in many cases of traumatic injury. There is conflicting evidence on the impact of alcohol use at the time of physical trauma on severity of injury and hospital course. Similarly, the significance of alcohol use disorder on outcomes in hospitalized trauma patients is unclear. This scoping review aims to provide a concise overview of the current literature surrounding peri-trauma alcohol use and alcohol use disorder on injury severity, in-hospital complications, patient outcomes, and long-term health impact of alcohol use in trauma. We will also explore the associated healthcare costs of this patient population.

### Methods

A systematic search of the following databases MEDLINE, EMBASE, and Cochrane Library will be completed to extract all studies that meet our inclusion criteria from January 2000 onwards. Case reports will be excluded. Two reviewers will screen all citations, abstracts, and full text articles. A third reviewer will act as tiebreaker at each stage of the screening process. A narrative synthesis without meta-analysis will be conducted and assessed based on the Preferred Reporting Items for Systematic Reviews and Meta-Analyses extension for Scoping Reviews (PRISMA-ScR) guidelines.

### Discussion

This review will contribute to the literature by providing a concise overview of the current data on the impact of alcohol on outcomes following trauma. We will explore the overall themes in the literature, limitations, and future directions to focus forthcoming research in this patient population.

### Scoping review registration

This project is registered via the Open Science Framework. The public registration is uniquely identified with the following DOI: https://doi.org/10.17605/OSF.IO/Z84WK.

**Data Availability Statement:** No datasets were generated or analysed during the current study. All relevant data from this study will be made available upon study completion.

**Funding:** The author(s) received no specific funding for this work.

**Competing interests:** The authors have declared that no competing interests exist.

## Support

There were no funders or sponsors involved in the development of this protocol.

## Introduction

Injuries are defined as "the transfer of energy to human beings at rates and in amounts above or below the tolerance for human tissue", and are considered unintentional (without the intent to harm) or intentional [1]. Intentional and unintentional injuries were the 8[th] leading cause of hospitalization in Canada in the 2018/19 fiscal year and accounted for $29.4 billion or $80 million per day of healthcare costs as of 2021 [2]. Furthermore, these injuries are the third leading cause of death for Canadians, with the annual mortality rate rising from 27.4 per 100 000 population in 2001 to 47.2 per 100 000 in 2021 [2]. Traumatic injuries, being tissue injury caused suddenly due to violence or accident, were responsible for 64% of injury-related deaths and 78% of hospitalizations [3, 4]. Rates of alcohol-associated presentation to the emergency department (ED), defined as self-reported alcohol use and/or elevated blood alcohol concentration (BAC), have been reported to be as high as 20–47% in the trauma population [5–7]. Alcohol has been so widely accepted as a significant reversible contributing factor to trauma that, in 2006, the American College of Surgeons Committee on Trauma introduced screening for problematic drinking as a requirement for designation as a level 1 or 2 trauma center and requires that all level 1 trauma centers provide an intervention for problem drinking [8].

Despite this apparent importance, the literature on the impact of alcohol on outcomes following trauma is limited and often contradictory. Some studies demonstrate that alcohol intoxication has a protective effect on injury; reporting decreased Injury Severity Scores (ISS), decreased mortality, shorter hospital length of stay (LOS), and fewer complications [9]. Conversely, other studies report increased ISS, increased rates of Intensive Care Unit (ICU) and ventilator days, and lower likelihood of discharge to home [10–12]. The heterogeneity of literature on the topic is likely due to the difficulty of conducting high quality research on this patient population. Most literature is based on the retrospective review of trauma registries and is therefore subject to the limitations associated with this type of study design including heterogeneity in definitions such as alcohol misuse; reliability of comprehensive records; differing study aims and outcomes; and confounding factors such as comorbidities.

A preliminary search of MEDLINE, the Cochrane Database of Systematic Reviews and JBI Evidence Synthesis was conducted and there were no systematic or scoping reviews completed or underway aside from this current study being undertaken by the authors.

Our study aims to provide a concise overview of the current literature surrounding alcohol exposure at the time of traumatic injury and alcohol use disorder on injury severity, in-hospital complications and mortality. We will also explore other outcomes including characterizing at-risk populations, the utility of brief motivational interviewing during hospital admission, long-term effects of alcohol use in trauma, and the financial burden on the healthcare system.

## Methods

### Protocol and registration

This protocol has been registered with Open Science Framework (https://doi.org/10.17605/OSF.IO/Z84WK) and is being reported in accordance with the Preferred Reporting Items for Systematic Reviews and Meta-Analyses extension for Scoping Reviews (PRISMA-ScR)

statement [13]. This scoping review conforms to the requirements of JBI scoping reviews. Any amendments to this protocol will be documented and published alongside the results of the scoping review.

## Eligibility criteria

**Inclusion criteria.** Our patient population will include adult patients, defined as age 15 years or older, who were admitted to a hospital for traumatic injuries occurring in the setting of acute alcohol intoxication or those who have a history of alcohol use disorder. The age limitation was decided based on the minimum legal drinking age of European countries included in the literature. Studies published since the year 2000 in English language from any country will be included, which is in keeping with similar evaluations of literature and functions to focus on modern literature due to evolution of practices in traumatology over time [14, 15]. Experimental and quasi-experimental study designs including randomized controlled trials, prospective and retrospective cohort studies, case-control, cross-sectional studies, mixed methodology, and descriptive observational/ qualitative studies will be selected from peer reviewed databases. Systematic reviews were excluded to focus on primary sources of evidence and to avoid influence from their conclusions. Unpublished works, grey literature, case reports and abstract articles were excluded so as to focus only on studies expected to possess a high degree of scientific rigor.

**Exclusion criteria.** Case reports, abstract only, or studies with full text articles that could not be accessed by the authors through the institutional access provided by the University of Manitoba will be excluded. Patients with traumatic brain injury as their only injury will be excluded because there already exists a large database of literature including systematic reviews and meta-analyses on this specific patient population [16, 17]. Literature with a study population consisting exclusively of the following cohorts will be excluded: military personnel, burn patients, substance use without concomitant alcohol use, and self-harm. These populations are often studied in isolation and the conclusions from these studies are not generalizable to the population presenting to trauma hospitals globally and therefore inclusion for the purpose of this scoping review may skew the results. In addition, study populations comprising exclusively of sexual assault victims and victims of psychological trauma are being excluded as our focus is on physical injuries and inclusion of these population may result in bias. Studies in which all patients died pre-hospital or in the ED will be excluded as the aim of this study is to examine the consequence of alcohol use on clinical outcomes throughout a hospital stay to identify intervenable measures. Patients who were not admitted to the hospital, those with only psychological trauma without physical trauma, or where data was collected from police reports or autopsy data, will be excluded as they typically lack data regarding key clinical outcomes evaluated in our study. Similarly, we will exclude studies that do not report on at least one of the outcome measures below. Inclusion and exclusion criteria are summarized in Table 1.

## Outcome measures

We will evaluate studies with at least one of the following clinical outcomes: mortality; Injury Severity Score; surgery because of injury; in-hospital complications; resource use; long term functional impact; and psychological or abstinence-based interventions. This selection is in keeping with outcomes encountered in existing trauma literature [18–20]. In-hospital complications are defined as harmful events or negative outcomes occurring during inpatient hospitalization that occur during the processes of care and treatment rather than the natural progression of diseases as well as any complication that is listed as such by the respective

Table 1. Scoping review inclusion and exclusion criteria.

| Inclusion Criteria | Exclusion Criteria |
|---|---|
| • Published in English language<br>• Age ≥ 15 years<br>• Publication year ≥ 2000<br>• Admitted to hospital due to a traumatic injury<br>• Injury associated with acute alcohol intoxication or history of alcohol use disorder. | • Case reports<br>• Studies where full text cannot be accessed<br>• Patients not admitted to hospital<br>• Police reports only<br>• Autopsy data only<br>• Patient population that consists exclusively of*:<br> ○ Military personnel<br> ○ Sexual assault victims<br> ○ Psychological trauma<br> ○ Burn patients<br> ○ Non-alcohol related substance use<br> ○ Self-harm<br> ○ TBI patients |

*Note that studies will be included if the patient categories comprise only part of the population

publication [18, 20]. These can include but are not limited to: cardiac arrest; arrythmia; sepsis; multiorgan failure; pneumonia; respiratory distress; pulmonary embolism; seizures; delirium; and hepatic failure [18, 19]. Demographic information such as, mean age, gender, ethnicity, socioeconomic status, injury type and mechanism, will be included. Resource use will be defined as any of the following: hospital length of stay (LOS); intensive care unit (ICU) LOS; healthcare costs; and propensity to leave against medical advice [18]. Long term functional impact is defined as any of the following: trauma recidivism; discharge disposition; likelihood of return to work; financial burden; and psychological or substance use disorders after admission [21]. Preventative measures taken at the time of admission for prevention of psychological harm, risk reduction, and mitigation of substance abuse will also be discussed [22].

## Search strategy

A structured search of the following electronic databases (from January 2000 onwards) will be undertaken including MEDLINE, Cochrane Library, EMBASE via OVID, SCOPUS, and CINAHL. The following keywords will be used: (injuries or wounds or trauma or traumas or wound or contusion or bruises or accident or MVC or penetrating or blunt or GSW) AND (alcohol* or binge drinking* or substance abuse* or ETOH or BAC or blood alcohol level or intoxication or substance misuse* or AWS or ethanol abuse) AND (acute care or level one trauma or trauma centre or trauma center or hospitaliz* or emergency department or emergency ward or ER or emergency room or emergency unit or intensive care unit or ICU or in-hospital). This search strategy was developed in consultation with a University of Manitoba librarian.

## Data selection and screening process

Covidence will be used to manage records and data throughout the review [14]. Duplicates will be removed using this software. Following a pilot test of ten percent of the database, abstracts will be independently screened by two reviewers (DL, CG) for assessment against the inclusion criteria. In cases where decisions for inclusion/exclusion are discordant, the principal investigator will act as the tiebreaker (JN). Full-text copies of the studies will be retrieved and independently reviewed by two authors (DL, CG) to ensure inclusion/exclusion criteria are met. Excluded articles or articles requiring further consensus will be reviewed and

| Author | Study Name | Year | Population & Study Design | Objective | Results | Conclusions & Odds ratio | Limitations |
|---|---|---|---|---|---|---|---|
| L Moore, HT Stelfox, A Boutin, et al. | Trauma centre performance indicators for nonfatal outcomes: A scoping review of the literature | 2013 | Scoping review Human studies from high income countries from earliest date to July 2011 | To evaluate the level of research on nonfatal outcomes in trauma | 14 non fatal outcomes identified from 40 included studies. These include adverse events, missed injuries, reintubation, ICU days, unplanned surgeries, HLOS, readmissions, functional capacity/ ADLs. | Adverse events and resource use were frequently used to evaluate trauma care, readmissions and function in daily activities were rarely used, and quality of life was never used. | Population limited to North America, Europe, & Australia. Studies may have been missed in data collection. Missing information in publications. |

**Fig 1. Data extraction table.**

discussed with the principal investigator (JN). Reasons for the exclusion of sources of evidence will be recorded in full in the final scoping review and presented in a PRISMA-ScR flow diagram.

## Data extraction, evaluation, and synthesis

A data extraction table will be designed to record relevant information from each study including: author; study; year of publication; population; study design; objective; results; conclusions; limitations; and odds ratios for quantitative outcomes (Fig 1). This table will be piloted by two reviewers (DL, CG) using ten percent of the full text articles and reviewed with the research team before proceeding with the rest of the data extraction. Any inconsistencies in data extraction or changes required to the charting table between reviewers will be discussed and resolved prior to proceeding to ensure accuracy and cohesiveness between data collectors. The remainder of the data extraction will be done and recorded independently by one of the two reviewers conducting the initial reviews (DL, CG). Following data extraction, the table will be reorganized, and articles will be grouped based on similar study designs and outcomes. The completed table will be included as an appendix to our final scoping review.

Due to the expected variability of the reporting of outcomes, a meta-analysis will not be performed. The data will instead be analyzed based on Synthesis Without Meta-analysis (SWiM) and Enhancing transparency in reporting the synthesis of qualitative research (ENTREQ) guidelines [23, 24]. We will summarize findings for each outcome in narrative format. Where appropriate a Grading of Recommendations Assessment, Development and Evaluation (GRADE) score will be applied to the summary of outcomes [25]. Due to the nature of a scoping review and expected lack of high-quality studies (prospective studies, randomized controlled trials) there will be no formal risk of bias assessment however limitations to included studies will be recorded on an outcome level within the discussion and study level within the data extraction table.

## Discussion

This review is designed to evaluate the current body of literature on alcohol involvement in outcomes after trauma. We expect it will also provide information about the impact of alcohol on at-risk demographics, healthcare utilization, discharge disposition, and the physical and psychological impact of trauma. Individual studies may be limited by small patient populations, single institutional data, risk of bias, and primarily retrospective study designs making it difficult to come to conclusions. The intention of our scoping review is to observe the trends and limitations in the literature for this patient population, identify gaps in the existing literature and inform future work on this important and highly relevant patient population.

## Supporting information

**S1 Checklist. PRISMA for systematic review protocols checklist.**
(DOCX)

## Acknowledgments

We would like to thank University of Manitoba librarian Carole Cook for assisting with our literature search strategy as well as research assistant Brenda Comaskey for reviewing and editing this protocol article.

## Author Contributions

**Conceptualization:** Nhayan Abdulla, Rae Spiwak, Sarvesh Logsetty, Jordan Nantais.

**Formal analysis:** Jordan Nantais.

**Investigation:** Chantae Garland, Donghyun Lee.

**Methodology:** Chantae Garland, Donghyun Lee, Jordan Nantais.

**Supervision:** Rae Spiwak, Sarvesh Logsetty, Jordan Nantais.

**Validation:** Jordan Nantais.

**Writing – original draft:** Chantae Garland, Nhayan Abdulla, Donghyun Lee, Jordan Nantais.

**Writing – review & editing:** Chantae Garland, Nhayan Abdulla, Donghyun Lee, Rae Spiwak, Sarvesh Logsetty, Jordan Nantais.

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
