## [Decision Letter · Decision Letter 0]

24 Jun 2024

PONE-D-24-08654The Impact of Alcohol Misuse in Trauma Patients: A Scoping Review ProtocolPLOS ONE

Dear Dr. Nantais,

Thank you for submitting your manuscript to PLOS ONE. After careful consideration, we feel that it has merit but does not fully meet PLOS ONE’s publication criteria as it currently stands. Therefore, we invite you to submit a revised version of the manuscript that addresses the points raised during the review process.

**This is an interesting protocol fro a scoping review to explore a potential connection between Alcohol use disorder and Trauma. There are certain concerns raised by both reviewers, with Reviewer 2 doing a more extensive review. Kindly address these concerns raised. **==============================

We look forward to receiving your revised manuscript.

Kind regards,

Lakshit Jain, MD

Academic Editor

PLOS ONE

Reviewers' comments:

Reviewer's Responses to Questions

**Comments to the Author**

1. Does the manuscript provide a valid rationale for the proposed study, with clearly identified and justified research questions?

Reviewer #1: Yes

Reviewer #2: Yes

2. Is the protocol technically sound and planned in a manner that will lead to a meaningful outcome and allow testing the stated hypotheses?

Reviewer #1: Yes

Reviewer #2: No

3. Is the methodology feasible and described in sufficient detail to allow the work to be replicable?

Reviewer #1: Yes

Reviewer #2: No

4. Have the authors described where all data underlying the findings will be made available when the study is complete?

Reviewer #1: No

Reviewer #2: No

5. Is the manuscript presented in an intelligible fashion and written in standard English?

Reviewer #1: Yes

Reviewer #2: Yes

6. Review Comments to the Author

You may also provide optional suggestions and comments to authors that they might find helpful in planning their study.

Reviewer #1: The paper describes a protocol for a scoping review on the impact of alcohol misuse in trauma patients. The protocol is well written and clear on processes, following PRISMA guidelines.

My suggestions are to consider adding HRQOL to your outcome measures, and secondly to consider adding primary care settings to your inclusion criteria and search strategy - or if not, to specify above that the review will be specifically for hospital patients.

Reviewer #2: Does the manuscript provide a valid rationale for the proposed study, with clearly identified and justified research questions?- Yes

Is the protocol technically sound and planned in a manner that will lead to a meaningful outcome and allow testing the stated hypotheses?- No

o The main concern here is the studies that are stated to be included. Studies to be included are stated on line 95-97. Please consider including additional studies like mixed methodology or qualitative pieces as they could provide additional information about ETOH exposure at the event, more so than strictly quantitative studies could. My concern is that including only quantitative studies will not sufficiently answer your research question.

o Further reference to if published, unpublished and/or grey literature are under consideration will help strengthen the protocol.

o Stating principles that are being followed would strengthen this protocol. Saying if JBI, Cochraine, etc are being followed and then applying associated principles would elevate this manuscript.

Is the methodology feasible and described in sufficient detail to allow the work to be replicable?- No

- More detail, and justification for study methodologies around replication is required and highlighted in the reviewer letter.

- The study itself would be feasible.

Have the authors described where all data underlying the findings will be made available when the study is complete?- No

No, however this is not applicable to a scoping review protocol.

Is the manuscript presented in an intelligible fashion and written in standard English?- Yes

7. PLOS authors have the option to publish the peer review history of their article (what does this mean?). If published, this will include your full peer review and any attached files.

Reviewer #1: No

Reviewer #2: No

---

## [Author Response · Author response to Decision Letter 0]

8 Aug 2024

Abstract:

- Please state the Open Science Framework study registration number.

o Thank you for the feedback. To our knowledge, there is no protocol registration number associated with protocols registered in the Open Science Framework. They do assign a unique DOI when the protocol is registered. We have clarified this and have provided the DOI number on lines 45-47.

Introduction:

- Providing a definition of unintentional injuries and traumatic injuries will help with reader comprehension. 

o Thank you for the comment. We have clarified the definitions of injuries (unintentional and intentional) and traumatic injury in order to provide further understanding of our focus. The definitions and associated references are provided in lines 54-56 and 60-61. 

- Line 74 states “no current or underway systematic reviews or scoping reviews on the topic were identified”. This is in a way contradictory as a scoping review is considered to be a type of systematic review as per JBI methodology. Please reword this line. 

o Thank you for the feedback. This sentence has been reworded on lines 78-79 to clarify this.

Methods:

- The method section is lacking as no scoping review principles are stated to be followed. To improve this section please state what scoping review principles you are following is it JBI, Cochraine, something else? Stating this and following the principles of your choice will help improve your manuscript.

o Thank you for this comment, we have clarified that we are following the PRISMA SCR as per the JBI template in the design and conduct of our review and protocol on line 90 -91.

- Line 84, please provide Open Science Framework registration link, please state the registration number instead. 

o Thank you for the feedback, please see above explanation regarding this. 

- Under eligibility criteria, please clarify in the methods section the specific types of literature being included in the scoping review. Will you be including published, unpublished and/or grey literature. Subsequently, please provide rational for which you are using and which you are excluding. 

o Thank you for the feedback. This has been now addressed in lines 102-107. Systematic reviews were excluded to focus on primary sources and unpublished/grey literature has been excluded as they are not expected to possess the same degree of scientific rigor and are not peer-reviewed. 

- Under inclusion criteria:

o On line 93-94 it is stated that studies from 2000 onward will be included with the justification “in keeping with similar evaluations of literature”. Please expand on this justification. Often scoping reviews do not restrict by dates, it is okay to do so but requires more justification.

Thank you for this comment. We have included further justification and references of literature in trauma that also focus on the year 2000 and beyond (lines 99-101). Primarily, we would like to focus on modern trauma care given its ongoing evolution.

o Studies to be included are stated on line 95-97. Please consider including additional studies like mixed methodology or qualitative pieces as they could provide additional information about ETOH exposure at the event, more so than strictly quantitative studies could. 

Thank you for this comment. We have further clarified this in line 102-104 to be more comprehensive. 

o Line 94 states that included studies will be included if they are published in English. This is common however groups such as JBI are moving away from English only and encouraging authors to include studies of all languages. As general advice from this reviewer please consider using free software such as DeepL (https://www.deepl.com/en/translator) to be able to include non-English studies. 

Thank you for the insightful comment. Although automated translational services are becoming more commonplace in scientific literature, we remain somewhat unsure as to whether the conclusions drawn from such translations may be biased by inaccuracies. Since English is the dominant language of the available literature, at this time it is our preference to retain this exclusion criteria.

- Under exclusion criteria:

o Please be explicit about where you are accessing articles through, for example line 99 states “…full text articles that could not be accessed will be excluded” but where is access coming from, an institution?

Thank you for this comment. We clarified the exclusion criteria to say full text articles that cannot be accessed by the authors through our university access (University of Manitoba) in line 109

o Line 102-104 states studies with exclusive groups that will not be included in the scoping review. Two groups are referenced that may cause the reader some confusion, the exclusion of cohorts with psychological trauma and burn patients. Please justify the exclusion of these groups.

Thank you for this comment. We separated the exclusion criteria in two sentences to be clearer in the reasoning of our exclusion criteria. These are listed in lines 113-118. We also reformatted Table 1 to also help clarify this issue (line 126). 

o Line 106 states “patients who were not admitted to the hospital… will be excluded”. Please clarify if patients who died in the ED will be included or excluded in this study. 

Thank you for this feedback. We felt that the outcomes examined required an assessment of an associated hospital admission, and that death very early in the course of treatment in an entire population would limit our ability to evaluate this. However, studies wherein only some patients died in the emergency department would still be included. We have clarified this in line 119-120.

o Table 1 is a great way to summarize inclusion and exclusion criteria for the reader.

In this table, under exclusion criteria TBI as only injury is excluded, but has the author considered what will be done with studies that report multiple cases where patients with TBI may be included. Please refine table to reflect if just studies with TBI only patients will be included or if they will be included in the before mentioned circumstances. 

Thank you for this feedback, we have made amendments to the table to clarify that any of the excluded patient groups will not result in exclusion of the study if they only comprise part of the population.

- Under outcome measures

o Line 122 states “Demographics when available will be evaluated (mean, age, etc…)” please reword this line to improve understanding. 

o Thank you for this comment, this has been clarified in lines 137-138.

- Under search strategy

o Later in the acknowledgements the authors state that a librarian was consulted. Please reference the consultation of a librarian under search strategy. 

o Thank you for this feedback. This has been clarified in lines 153-154.

o It may strengthen the scoping review to have the search strategy Peer Review of Electronic Search Strategies (PRESSED). Here an additional librarian peer reviews the search strategy. If this has not been done, please considering having the strategy PRESSED. If this was completed already, please state it explicitly. Thank you for this feedback. We did reach out to our local University of Manitoba Library however unfortunately we were unsuccessful at having this reviewed in the timeframe given institutional restraints. 

o Line 138-139 reference when the search will be completed. Please reword and remove mention of submission of the manuscript. Instead it would be sufficient to state the month and year the search was carried out. 

o Thank you for this feedback. We have ensured only the month and year of when the search was completed was included.

- Under data extraction, evaluation and synthesis

o Throughout this section “authors” are referenced, please change the word to reviewers. 

o Thank you for this feedback. This has been completed. 

o It is great practice to pilot as the author referenced. However this section would be strengthened if the authors stated how many articles were piloted, instead of “ten percent of the full text articles”. This would be quite a lot depending on how many articles are yielded. I would encourage the authors to pick a number instead of a percentage to pilot. For example ten articles could be a sufficient pilot. 

o Thank you for this feedback. At the time of receiving this review, the pilot has been completed which far exceeded the suggested 10 articles. 

o Line 164 references qualitative studies, however in previous section they are not mentioned as being included. Please clarify. 

o Thank you for this comment. This has been addressed in line 103. 

o Line 165 states “few will summarize findings for each outcome reported by at least 2-3 papers in a narrative format”. Please provide rational for the 2-3 paper mention, or select a definitive number with justification. 

o Thank you for this comment. We have clarified this issue in lines 179-180. We have decided to comment on all qualifying articles for completeness.

o Line 168 please edit from “lack of high quality” to “lack of high quality studies’. 

o Thank you for this feedback. This has been done on line 182-183.

Discussion:

- Line 178 references “strong evidence-based conclusions”. Please be aware than coming to definitive evidence-based conclusion is not the purpose of a scoping review, instead it is a scan of the available literature. Please remove this reference. 

- Thank you for this feedback. This has been completed.

- Line 179 please change “trend” to “trends”. 

- Thank you for this feedback. This has been completed on line 192. 

Other:

- In acknowledgements the use of a librarian is reference, as previously mentioned please address this consultation under search strategy.

- Thank you for this comment. This has been addressed in line 153-154.

- In acknowledgements the help of a research assistant is referenced for reviewing and editing the protocol article. Please reference authorship guidelines of PLoS One and reflect on Brenda’s contributions. Please consider if the research assistants contribution would be substantial enough to be an author on this paper. 

- Thank you for this feedback. Although Brenda aided in editing and reviewing the protocol article, she did not contribute to the overall creation of the article. After consideration and review of the authorship guidelines we feel that inclusion of Brenda in the acknowledgments is most appropriate.

- I would encourage the authors to reflect on their choice of a scoping review. As presented, with the above edits applied a scoping review would be appropriate for this research problem. However, throughout the manuscript at times it does read as if the authors really want to do a traditional scoping review without a meta analysis. 

- Thank you for this feedback. All the authors are in agreeance that a scoping review would be much better suited in this circumstance. The main reason being that given our preliminary evaluation of the literature we think that it is very unlikely that there will be suitable high-quality evidence to perform a systematic review with or without meta-analysis. In large part the literature is limited by inconsistent outcome definitions and low-quality evidence, greatly decreasing our ability to draw meaningful conclusions. We hope that our scoping review will function to sweep across the databases to identify if the evidence is of suitable quality for a deeper analysis, and failing that, where gaps may be addressed with further primary literature.

---

## [Editor Report · Decision Letter 1]

12 Aug 2024

The Impact of Alcohol Misuse in Trauma Patients: A Scoping Review Protocol

PONE-D-24-08654R1

Dear Dr. Nantais,

We’re pleased to inform you that your manuscript has been judged scientifically suitable for publication and will be formally accepted for publication once it meets all outstanding technical requirements.

Kind regards,

Lakshit Jain, MD

Academic Editor

PLOS ONE